

# Effects of pharmaceutically active compounds (PhACs) on fish body and scale shape in natural waters

Adam Staszny[1], Peter Dobosy[2], Gabor Maasz[3,4], Zoltan Szalai[5,6], Gergely Jakab[5,6,7], Zsolt Pirger[3], Jozsef Szeberenyi[5], Eva Molnar[3], Lilianna Olimpia Pap[1], Vera Juhasz[1], Andras Weiperth[1], Bela Urbanyi[8], Attila Csaba Kondor[5] and Arpad Ferincz[1]

[1] Department of Freshwater Fish Ecology, Institute of Aquaculture and Environmental Safety, Hungarian University of Agriculture and Life Sciences, Gödöllő, Hungary
[2] Danube Research Institute, MTA-Centre for Ecological Research, Budapest, Hungary
[3] Balaton Limnological Institute, MTA-Centre for Ecological Research, Tihany, Hungary
[4] Soós Ernő Research and Development Center, University of Pannonia, Nagykanizsa, Hungary
[5] Geographical Institute, Research Centre for Astronomy and Earth Sciences, MTA Centre for Excellence, Budapest, Hungary
[6] Department of Environmental and Landscape Geography, Eötvös Loránd University, Budapest, Hungary
[7] Institute of Geography and Geoinformatics, University of Miskolc, Miskolc, Hungary
[8] Institute of Aquaculture and Environmental Safety, Hungarian University of Agriculture and Life Sciences, Gödöllő, Hungary

Corresponding author
Adam Staszny,
Staszny.Adam@uni-mate.hu

## ABSTRACT

**Background:** In recent years, there are growing concerns about pharmaceutically active compounds (PhACs) in natural ecosystems. These compounds have been found in natural waters and in fish tissues worldwide. Regarding their growing distribution and abundance, it is becoming clear that traditionally used risk assessment methodologies and ecotoxicological studies have limitations in several respects. In our study a new, combined approach of environmental impact assesment of PhACs has been used.

**Methods:** In this study, the constant watercourses of the suburban region of the Hungarian capital (Budapest) were sampled, and the body shape and scale shape of three fish species (roach *Rutilus rutilus*, chub *Squalius cephalus*, gibel carp *Carassius gibelio*) found in these waters were analyzed, based on landmark-based geometric morphometric methods. Possible connections were made between the differences in body shape and scale shape, and abiotic environmental variables (local- and landscape-scale) and measured PhACs.

**Results:** Significant connections were found between shape and PhACs concentrations in several cases. Despite the relatively large number of compounds (54) detected, citalopram, propranolol, codeine and trimetazidine significantly affected only fish body and scale shape, based on their concentrations. These four PhACs were shown to be high (citalopram), medium (propranolol and codeine), and low (trimetazidine) risk levels during the environmental risk assessment, which were based on Risk Quotient calculation. Furthermore, seven PhACs (diclofenac, Estrone (E1), tramadol, caffeine 17α-Ethinylestradiol (EE2), 17α-Estradiol (aE2), Estriol (E3)) were also categorized with a high risk level. However, our morphological studies indicated that only citalopram was found to affect fish phenotype amongst

the PhACs posing high risk. Therefore, our results revealed that the output of (traditional) environmental/ecological risk assessment based on ecotoxicological data of different aquatic organisms not necessarily show consistency with a "real-life" situation; furthermore, the morphological investigations may also be a good sub-lethal endpoint in ecotoxicological assessments.

# INTRODUCTION

The first detection of pharmaceutically active compounds (PhACs) in aquatic ecosystems and drinking water dates back to the 1980s (*Richardson & Bowron, 1985*; *Watts et al., 1983*). Since then, an emerging number of studies have reported the distribution and the potential threat posed by these compounds (*Boxall et al., 2012*; *Datel & Hrabankova, 2020*; *Dietrich, Webb & Petry, 2002*). Selective Serotonin Reuptake Inhibitors (SSRIs), β-blockers and anti-inflammatories are considered to be the most abundant drug residuals occurring in surface waters (*Boxall et al., 2012*). These compounds can be released into natural waters via several ways. The main sources of pollution are Wastewater Treatment Plants (WWTPs) (after the excretion of human waste) (*Subedi et al., 2012*), the pharmaceutical industries and the excretion of drugs from animals used in agriculture (*Boxall et al., 2012*). The recent technologies of WWTPs cannot eliminate these compounds fully from wastewater (*Golet et al., 2001*; *Ternes et al., 1998*; *Tsui et al., 2014*; *Yang et al., 2020*). To minimize the potential environmental risk posed by PhACs, several regulations for ecotoxicological testing have been enacted (*EMEA, 2006*). In recent years, several weaknesses of these regulations have been reported in scientific articles (*Ankley et al., 2007*; *Boxall et al., 2012*) such as: (1) official tests usually use lethal endpoints, (2) little attention is paid to metabolites, (3) different regulations for human and for veterinary drugs, (4) tests for unique agents, (5) calculating the degradation of compounds and (6) overabundant compounds (over 4.000 drug substances) to test all of them. These weaknesses and the resulting shortcomings in risk assessment procedures may cause uncertainties regarding their validity. If these points are not addressed and alternative, more adequate risk assessment techniques would not added to the regulations, then a false illusion of low risk may result in many cases. Therefore, the current shortcomings need to be examined in detail in order to better understand the problem. It is a well-known fact that several biotic and abiotic factors can influence the body shape of fish, such as food availability (*Currens et al., 1989*; *Marcil, Swain & Hutchings, 2006*; *Park et al., 2001*), food type (*Day, Pritchard & Schluter, 1994*), temperature (*Beacham, 1990*; *Šumer et al., 2005*), and the presence or absence of predators (*Brönmark & Miner, 1992*). In addition, it has also been proven that environmental parameters can affect the shape of fish scales (*Ibáñez, 2015*; *Staszny et al., 2013*; *Takács et al., 2016*). The effect of basic chemical parameters (e.g., ion concentrations) of the water may also affect the phenotype of fish, however their effect on the shape (body or scale) is unclarified
(*Çoban et al., 2013*; *Franklin et al., 2005*; *Schlenk & Benson, 2001*). Due to the chronic, multigenerational exposure of fishes to PhACs, phenotypic alterations are possible, and there is evidence that progestogen contaminations can affect somatic indices (*Maasz et al., 2017*). Therefore, the aim of this study was (1) to find connections between the PhACs measured in small watercourses and the body and scale shape of selected fish species; and (2) to describe which type of PhACs or abiotic environmental factors are responsible for anatomical differences.

## MATERIALS AND METHODS

### Ethics statement

This study followed all relevant national and international guidelines concerning the care and welfare of fish (*Algers et al., 2009*; *Johansen et al., 2006*). Fish samplings were authorized by the Minister of Agriculture (Permit no.: HHgF/298-1/2016) and fish collection for laboratory examinations was authorized by the Government Office of Pest county (Permit no.: XIV-I-001/2302-4/2012). During sampling, an effort was made to minimize the suffering of fish and all fish were anaesthetized with a lethal dose of clove oil after collection. No endangered species (according to the IUCN Red List of Threatened Species v. 13 (www.iucnredlist.org) and National Law Protected (http://www.termeszetvedelem.hu/)) were caught during this study.

### Study area

The study was performed in the suburban area of Budapest, which is the capital and the biggest city in Hungary and in the Carpathian Basin. Altogether, 22 points were sampled for chemical analysis during 2017–2018, and 420 specimens of three species (140 roach *Rutilus rutilus*, 180 chub *Squalius cephalus*, 100 gibel carp *Carassius gibelio*) were collected in 20 sampling points from 10 streams during 29 sampling occasions (Fig. 1). Body-and scale-shape data of 20 specimens/sites were included in the analyses, the number of sampling sites, where the necessary number of specimens were available has been indicated in Table 1.

### Water sampling and chemical analysis

Water samples were taken during low water-level periods. General water chemical analysis was performed in the field (Hanna HI 98194 for dissolved $O_2$, electric conductivity, pH, total dissolved solids, temperature; Macherey-Nagel VisColor PF12 spectrophotometer for $NO_2^-$, $NO_3^-$, $NH_4^+$, $PO_4^{3-}$). For further laboratory analyses ($F^-$, $Cl^-$, $SO_4^{2-}$, $NO_2^-$, $NO_3^-$, $PO_4^{3-}$, $NH_4^+$, $Ca^{2+}$, $Mg^{2+}$, $Na^+$, $K^+$) samples were collected in 500-ml borosilicate glass containers. Samples for total organic carbon (TOC) measurements were taken in white, borosilicate containers (50 ml sample with 500 µl 2M hydrochloric acid (VWR International, Monroeville, PA, USA)). For the elemental analysis, a 10-ml water sample was filtered through a 0.45 µm diameter syringe filter, into polypropylene centrifuge pipes free from metal pollutants, and 100 µl NORMATOM nitric acid (VWR International, Monroeville, PA, USA) was added. TOC and total nitrogen (TN) concentrations were measured by using a Multi N/C 3100 TC-TN analyzer (Analytik Jena, Germany). For the

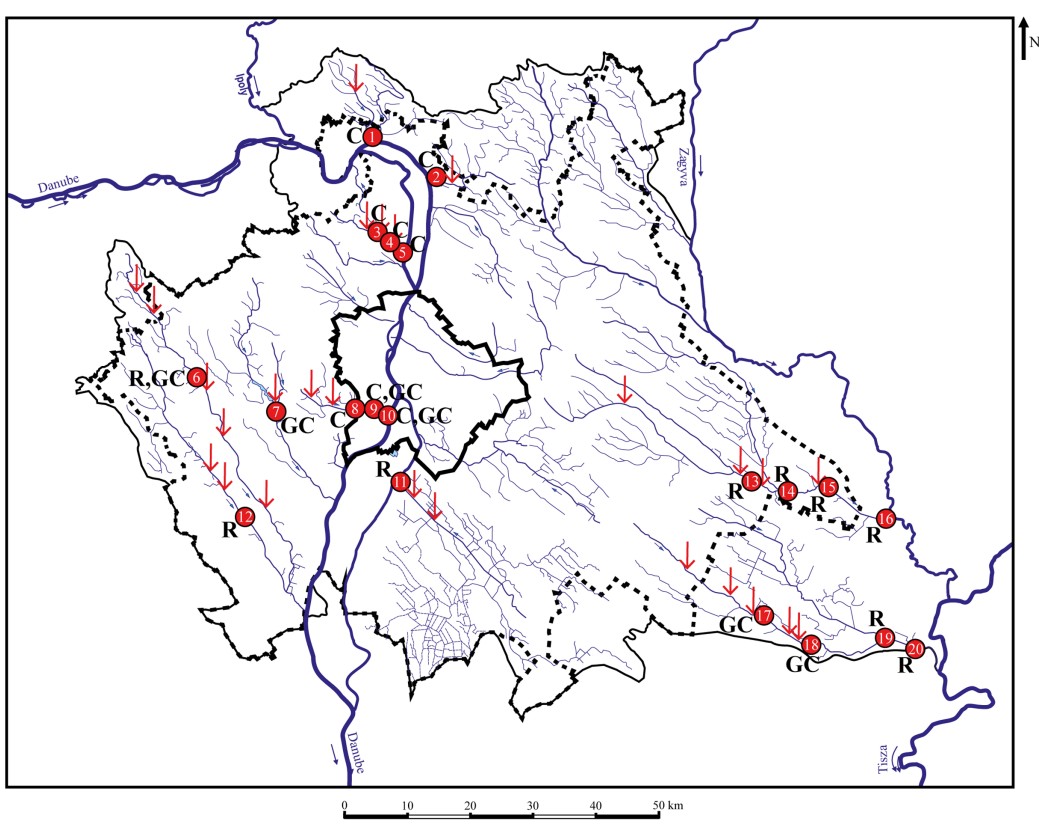

**Figure 1 Sampling points with sufficient individuals. Red vertical arrows shows WWTPs introductions.** C, chub; R, roach; GC, gibel carp; 1—MORVER; 2—GOMVAC; 3—BUKIZB; 4—BUKSZE; 5—BUKTOR; 6—SZEBIC; 7—BENBIA; 8—HOSKAM; 9—HOSKEL; 10—HOSTOR; 11—DTCDUN; 12—VALBAR; 13—TAPTAP; 14—TAPSZE; 15—TAPGYO; 16—TAPUJS; 17—GERCEG; 18—GERTOR; 19—GERKOR; 20—GERTOS. 

**Table 1 Number of sampled species and sampling points.**

| Fish species | No. of sampling points | No. of individuals/sampling points | Suitable data for analysis |
|---|---|---|---|
| Roach (*Rutilus rutilus*) | 6 | 20 | Scale |
| Roach (*Rutilus rutilus*) | 7 | 20 | Body |
| Chub (*Squalius cephalus*) | 9 | 20 | Scale |
| Chub (*Squalius cephalus*) | 6 | 20 | Body |
| Gibel carp (*Carassius gibelio*) | 5 | 20 | Scale |
| Gibel carp (*Carassius gibelio*) | 4 | 20 | Body |

determination of anions ($F^-$, $Cl^-$, $SO_4^{2-}$, $Br^-$, $NO_3^-$) and cations ($NH_4^+$, $Ca^{2+}$, $Mg^{2+}$, $Na^+$, $K^+$), a Dionex ICS 5000+ dual channel ion chromatograph (Thermo Fisher Scientific, Waltham, MA, USA) was used. $PO_4^{3-}$, $NO_2^-$ concentrations, alkalinity as well as total hardness were measured by standard titrimetric and spectrophotometric methods (*Eaton et al., 2005*). The concentration of heavy metals was determined by using PlasmaQuant MS Elite inductively coupled plasma mass-spectrometer (Analytik Jena, Germany).

For the PhACs measurements, a brown borosilicate glass container with Teflon faced caps (Thermo Fisher Scientific, Waltham, MA, USA) was filled with 2 l water sample, into which 2 ml of HPLC purity formic acid (VWR International, Monroeville, PA, USA) was added. The samples were immediately stored in 4 °C, and transported to the laboratory in a dark cooler box (Dometic CFX40W) within 4 hours, where they were then extracted.

Details of the sample preparation, extraction and analysis process for PhACs have also been described in our earlier papers (*Jakab et al., 2020*; *Kondor et al., 2020*; *Maasz et al., 2019*). Briefly, for sample quantification, the water samples were acidified with formic acid and spiked with the corresponding mass-labelled internal standard (IS). Because of the relatively low concentrations, analytes were isolated by an AutoTrace 280 automatic solid-phase extraction system (Thermo Fisher Scientific, Waltham, MA, USA) using Strata X-CW cartridges (#8B-S035-FCH, Phenomenex, Torrance, CA, USA). To reach the adequate sensitivity, dansyl-chloride was used in the derivatization of steroid agents. A supercritical fluid chromatography (ACQUITY UPC2 system, Waters) coupled with tandem mass spectrometry (MS/MS) (Xevo TQ-S Triple Quadrupole, Waters) was used to analyze and quantify the selected drug residues. Data were recorded by MassLynx software (V4.1 SCN950) in triplicates using TargetLynx XS software for evaluation. The compound separation was performed on an ACQUITY UPC2 BEH analytical column (#186007607, Waters) with 3.0 mm × 100.0 mm, 1.7 μm particle size.

## Fish sampling

Fish were caught by electrofishing, and all sampling was undertaken based on the EU Water Framework Directive (EU WFD) (*European Commission, 2009*) and Hungarian Biodiversity Monitoring System protocols (www.termeszetvedelem.hu). Sampled watercourse sections belonged to River1 (bed width under 5 m, water depth <1 m) and River2 (bed width over 5 m, water depth <2 m) categories, therefore a battery-powered electrofishing device (HANS-GRASSL IG200/2) was used, with a 150-m section length wading in the water upstream. Two watercourses belonged to the River3 (bed width under 30 m, water depth >2 m) category; therefore an aggregator-powered electrofishing device (HANS-GRASSL EL63II) was used, with a 300-m section length leading from a rubber boat going downstream. At every sampling point, 20 specimens comprised of common fish species (not endangered and not protected) were euthanized by using clove oil and stored at −20 °C.

## Environmental characterization of sampling sites

The most important environmental variables were recorded at two levels: local level and landscape level (Table 2). The two levels of environmental variables were analyzed separately.

## Morphometric analysis

For body morphometrics, after defrosting, a high resolution digital picture was taken of the left side of all specimens using a NIKON D7200 DSLR camera, with a AF-S NIKKOR 35 mm 1:1.8G objective, to avoid variability of side-effects (*Takács et al., 2018*). Standard

**Table 2 Local- and landscape-scale environmental variables used to characterize sampling points.**

| | Name | Abbreviation | Measure |
|---|---|---|---|
| Local environmental characteristics | Woody stemmed coastal vegetation within 1 m from riverbed | Wood 1 m | Shoreline coverage (%) |
| | Woody stemmed coastal vegetation within 10 m from riverbed | Wood 10 m | Shoreline coverage (%) |
| | Soft stemmed coastal vegetation within 1 m from riverbed | Soft 1 m | Shoreline coverage (%) |
| | Soft stemmed coastal vegetation within 10 m from riverbed | Soft 10 m | Shoreline coverage (%) |
| | Riverbed width | Width | m |
| | Water depth | Depth | cm |
| | Flow rate | Flow | m/s |
| | Sediment—detritus | Detritus | Bottom coverage (%) |
| | Sediment—mud | Mud | Bottom coverage (%) |
| | Sediment —sand | Sand | Bottom coverage (%) |
| | Sediment—gravel | Gravel | Bottom coverage (%) |
| | Sediment—stone | Stone | Bottom coverage (%) |
| | Bottom—rock | Rock | Bottom coverage (%) |
| | Bottom—concrete | Concrete | Bottom coverage (%) |
| | Macrophyte coverage | Macrophyte | Coverage (%) |
| Landscape-scale environmental characteristics | Catchment size over the sampling point | Catch.size | $km^2$ |
| | Inhabited area in the catchment | Inhab.area | $km^2$ |
| | Size of artificial surface in the catchment | Art.surface | $km^2$ |
| | Agricultural surface in the catchment | Agri.surface | $km^2$ |
| | Forest vegetation in the catchment | Forest | $km^2$ |
| | Non-forest vegetation in the catchment | Non-forest | $km^2$ |
| | Wetland area in the catchment | Wetland | $km^2$ |
| | Ponds above the sampling point | Ponds | number |
| | Distance from estuary | Distance | km |
| | Distance from the nearest known wastewater discharge | Wastewater.dis | km |
| | Altitude of sampling point | Altitude | m |
| | Average altitude of the catchment | Avg.altitude | m |

length and wet weight were measured with an accuracy of 1 mm and 0.1 g, respectively. Sex was determined by dissection, after the digital photo was captured (Table S1).

Five well-developed scales were removed from every individuals' left side from the flank. Scales were placed between glass slides and scanned using an upper-light scanner (EPSON Perfection V850 Pro) with high resolution (2,400 dpi). One scale per specimen was used for the analysis. Body and scale shape were analyzed using landmark-based geometric morphometry (*Zelditch et al., 2004*). Ten landmarks were placed on fish body and seven landmarks on fish scales (Fig. 2). For further multivariate analysis, we used the MorphoJ software package (*Klingenberg, 2011*). To derive shape variables from the raw landmark coordinates, a generalized least-squares Procrustes superimposition was applied to scale, translate and rotate the coordinates (*Rohlf, 1990*). To eliminate the variances associated with allometric growth, a regression analysis was performed between the logarithm of centroid sizes and the Procrustes coordinates. The regression residuals were used for
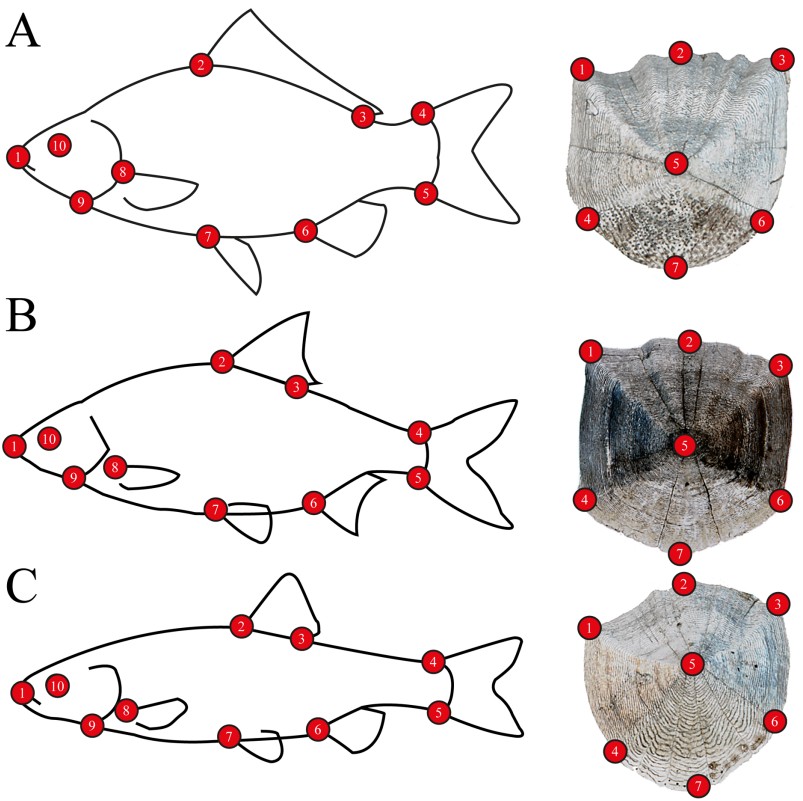

**Figure 2** Morphometric landmarks on (A) a schematic gibel carp (*Carassius gibelio*) and a gibel carp scale, (B) a schematic roach (*Rutilus rutilus*) and a roach scale, (C) a schematic chub (*Squalius cephalus*) and a chub scale.

further analysis (*Zelditch et al., 2004*). The Procrustes distance (*Pd*) was used in Canonical Variates Analysis (CVA) for computing group differences, and permutations tests with 1000 iterations were performed to test for significance.

## Ecological risk assessment

Ecological risk characterization for PhACs is usually performed by calculating and categorizing a risk quotient (RQ). RQ is a ratio of MEC/PNEC, in which PNEC (predicted no effect concentration) is the estimated highest concentration of an individual PhAC not affecting the aquatic ecosystem, and MEC is the maximum measured environmental concentration in the studied surface water. In general, RQ < 0.01 refers to a negligible risk, 0.01 < RQ < 0.1 denotes a low risk, 0.1 < RQ < 1 indicates a medium risk, while RQ > 1 represents a high risk to the aquatic ecosystem.

Predicted no effect concentration (PNEC) derives from the ratio of available ecotoxicological data (e.g., NOEC, EC50, LC50, HC5) and an assessment factor (AF). When the PNEC value was not available in the literature, we used a selected ecotoxicological data/AF quotient keeping in mind the priorities between the raw data (e.g., applying experimental results instead of extrapolated modelled data, and chronic outcomes in place of acute test results). The magnitude of the AF varies between 1,000 and 5, and it depends on the available ecotoxicological information. The uncertainty (i.e., AF) of the data decreases

with expanding of the relevant data set. If PNEC can be calculated only based on acute test results, then AF = 1,000. If PNEC can be derived from chronic data of a species, then AF = 100. Its value further decreases if ecotoxicological chronic test results are available at multiple different trophic levels: AF = 50 (two levels) or AF = 10 (three levels). If PNEC can be determined knowing of hazardous concentration for 5% of species investigated (HC5 based on ecotoxicological results of at least five species), then AF = 5. When data are available for each trophic level, the lowest concentration was selected to determine PNEC since environmental risk assessment is based on the most sensitive elements of the ecosystem (*Molnar, Maasz & Pirger, 2020*). PNECs with raw ecotoxicological data and AFs are presented in Table S2.

## Statistical analysis

Background variables were categorized into four groups: PhAC data, general water chemistry data, local environmental variables data and landscape-scale environmental variables. All variables were numeric and log10 transformed before further analyses. An unconstrained Principal Component Analysis conducted on the shape datasets ($x$ and $y$ coordinates of the regression residuals) was followed by the passive projection of the explanatory variables. The number of permutations in a Monte-Carlo simulation were set to 1,000. In the first model, body shape data, while in the second model, scale shape data, were used with all the environmental variables listed in the dataset. Where forward selection revealed significant effects, variance partitioning was used to assess the relative contribution of the different variable groups (*Borcard, Legendre & Dapeau, 1992*). Additional Mantel tests were performed on shape-variables (Mahalanobis and Procrustes distances) and PhACs concentrations, to assess the site-specific component of differences.

## RESULTS

### PhAC data from sampling points

Altogether 54 different types of PhACs were found in the water samples from the sampling points (Table 3). Three compounds were detected in a μg/l concentration range in examined samples, lamotrigine (maxMEC = 14 338.3 ng/l), caffeine (maxMEC = 13 635 ng/l), and diclofenac (maxMEC = 2 201.7 ng/l). The remaining 51 PhACs were measured in a few hundred, a few tens, or a few ng/l concentration ranges each above the limit of detection. A total of 27 PhACs were used in analysis based on their RQ-values, from which eight showed high, eight showed medium and the remaining eleven PhACs received a low risk classification based on the environmental risk assessment (Table 3). To perform the risk assessment using relevant ecotoxicological data, we used the AF and PNEC values of detected PhACs (see Table S2).

### Morphometric analysis

Significant differences were found between the average shape of fish stocks in all three species based on both fish body- and scale shape. In the case of roach body-shape, the

**Table 3 Measured Pharmaceutically Active Compounds (PhACs) from the water samples of sampling points.**

| PhACs | Abbreviation | LOQ ng/L | No. of sampling points found | maxMEC ng/L | PNEC | RQ | Risk level |
|---|---|---|---|---|---|---|---|
| **Diclofenac** | **DICL** | **0.5** | **20** | **2201.700** | **1.06E+01** | **207.708** | High risk |
| **Estrone** | **E1** | **0.05** | **20** | **38.161** | **1.00E+00** | **38.161** | |
| **Tramadol** | **TRAM** | **0.1** | **20** | **454.580** | **3.20E+01** | **14.206** | |
| **Caffeine** | **CAFF** | **10** | **20** | **13635** | **2.32E+03** | **5.877** | |
| **17α-ethinylestradiol** | **EE2** | **0.05** | **7** | **2.241** | **4.40E-01** | **5.093** | |
| **17α-estradiol** | **aE2** | **0.05** | **1** | **8.491** | **2.00E+00** | **4.245** | |
| **Estriol** | **E3** | **0.05** | **2** | **1.578** | **4.65E-01** | **3.394** | |
| *Citalopram* | *CITA* | *0.1* | *20* | *20.942* | *1.00E+01* | *2.094* | |
| **Theophylline** | **THEO** | **10** | **20** | **874.173** | **1.00E+03** | **0.874** | Medium risk |
| **Temazepam** | **TEMA** | **0.1** | **15** | **4.504** | **7.08E+00** | **0.636** | |
| **17β-estradiol** | **bE2** | **0.05** | **16** | **0.972** | **2.00E+00** | **0.486** | |
| **Metoclopramide** | **MCLO** | **0.2** | **15** | **23.626** | **5.60E+01** | **0.422** | |
| *Propranolol* | *PROP* | *0.1* | *20* | *14.870* | *4.11E+01* | *0.362* | |
| *Codeine* | *CODE* | *5* | *1* | *20.030* | *6.00E+01* | *0.334* | |
| **Clozapine** | **CLOZ** | **0.1** | **20** | **53.478** | **2.85E+02** | **0.188** | |
| **Trazodone** | **TRAZ** | **0.05** | **3** | **1.032** | **9.00E+00** | **0.115** | |
| **Losartan** | **LOSA** | **0.1** | **20** | **165.930** | **1.90E+03** | **0.087** | Low risk |
| **Carbamazepine** | **CARB** | **0.1** | **20** | **821.385** | **1.00E+04** | **0.082** | |
| **Propafenone** | **PROF** | **0.5** | **20** | **80.350** | **1.02E+03** | **0.079** | |
| **Ketamin** | **KETA** | **0.5** | **15** | **47.717** | **8.61E+02** | **0.055** | |
| **Lidocaine** | **LIDO** | **0.1** | **20** | **133.910** | **2.61E+03** | **0.051** | |
| **Bisoprolol** | **BISO** | **0.5** | **16** | **154.720** | **3.15E+03** | **0.049** | |
| **Alprazolam** | **ALP** | **0.1** | **20** | **20.561** | **5.08E+02** | **0.040** | |
| *Trimetazidine* | *TRIM* | *20* | *5* | *209.463* | *6.55E+03* | *0.032* | |
| **Tiapride** | **TIPA** | **0.1** | **20** | **177.606** | **8.72E+03** | **0.020** | |
| **Naproxen** | **NAPR** | **0.1** | **1** | **287.130** | **1.51E+04** | **0.019** | |
| **Midazolam** | **MIDA** | **0.1** | **5** | **4.371** | **2.89E+02** | **0.015** | |
| Paracetamol | PARA | 20 | 1 | 550.820 | 5.72E+04 | 0.010 | Negligible risk |
| Cocaine | COCA | 0.05 | 11 | 21.840 | 2.28E+03 | 0.010 | |
| Zolpidem | ZOLP | 0.01 | 18 | 4.384 | 5.19E+02 | 0.008 | |
| Bupropion | BUPR | 0.5 | 8 | 7.432 | 9.50E+02 | 0.008 | |
| Betaxolol | BET | 0.5 | 7 | 6.350 | 1.24E+03 | 0.005 | |
| Oxazepam | OXAZ | 0.1 | 11 | 5.581 | 1.92E+03 | 0.003 | |
| Metoprolol | MPRO | 0.1 | 20 | 150.161 | 6.15E+04 | 0.002 | |
| Nordiazepam | NORD | 0.1 | 9 | 2.750 | 1.19E+03 | 0.002 | |
| Mirtazapine | MIRT | 0.1 | 20 | 66.310 | 3.20E+04 | 0.002 | |
| Pethidine | PETH | 0.1 | 13 | 1.218 | 6.89E+02 | 0.002 | |
| Risperidone | RISP | 0.1 | 1 | 1.230 | 1.12E+03 | 0.001 | |
| Zopiclone | ZOPI | 0.1 | 1 | 2.750 | 4.75E+03 | 0.001 | |

*(Continued)*

| Table 3 (continued) | | | | | | | |
|---|---|---|---|---|---|---|---|
| PhACs | Abbreviation | LOQ ng/L | No. of sampling points found | maxMEC ng/L | PNEC | RQ | Risk level |
| Fentanyl | FENT | 0.1 | 2 | 0.307 | 5.39E+02 | 0.001 | |
| Olanzapine | OLAN | 5 | 13 | 54.071 | 1.41E+05 | $3.83 \times 10^{-4}$ | |
| Verapamil | VERA | 0.05 | 7 | 10.920 | 3.60E+04 | $3.03 \times 10^{-4}$ | |
| Perindopril | PERI | 0.1 | 20 | 285.461 | 9.90E+05 | $2.88 \times 10^{-4}$ | |
| Diazepam | DIAZ | 0.1 | 2 | 0.605 | 2.60E+03 | $2.33 \times 10^{-4}$ | |
| Carvedilol | CARV | 0.1 | 1 | 0.330 | 1.55E+03 | $2.12 \times 10^{-4}$ | |
| Ethylmorphine | EMOR | 0.5 | 12 | 15.869 | 1.33E+05 | $1.19 \times 10^{-4}$ | |
| Lamotrigine | LAMO | 5 | 20 | 14338.300 | 1.50E+08 | $9.56 \times 10^{-5}$ | |
| Quetiapine | QUET | 0.1 | 1 | 0.830 | 1.00E+04 | $8.30 \times 10^{-5}$ | |
| Warfarin | WARF | 0.1 | 3 | 0.880 | 1.20E+04 | $7.33 \times 10^{-5}$ | |
| Methadone | METH | 0.02 | 3 | 1.202 | 3.81E+04 | $3.15 \times 10^{-5}$ | |
| Benzoyl-ecgonine | BEC | 0.1 | 13 | 2.223 | 6.81E+06 | $3.26 \times 10^{-7}$ | |
| Cinolazepam | CINO | 0.1 | 20 | 394.197 | n.d. | n.d. | n.d. |
| Drospirenone | DROS | 1 | 2 | 2.999 | n.d. | n.d. | |
| Lacosamide | LACO | 0.5 | 18 | 82.549 | n.d. | n.d. | |

**Note:**
Compounds in bold were used in analysis based on their Risk Quotient (RQ), compounds in italics had a significant effect on fish shape, n.d., no data.

differences based on stream, as well as in scale shape (Fig. 3), significant differences and *Pd*-values are shown in Table 4 for body shape and Table 5 for scale shape.

Sampling points of Tápió Stream were discriminated from the others (Szent László Stream, Gerje Stream) along the first axis of CVA, according roach body shape. Significant differences were observed between GERTOS and every other points, based on Hotelling's *t*-test (Fig. 3; Table 4). SZEBIC has been differed significantly only from TAPTAP. Scale shape of roach proved to be different in TAPUJS, than most of other sites.

In the case of chub body- and scale shape, there were no clear connections found with the stream (Fig. 4); significant differences and *Pd*-values are shown in Table 6 for body shapes and Table 7 for scale shapes. Figure 4 suggests negative correlation between the distance from the estuary and CV2 (HOSTOR < HOSKEL < HOSKAM; BUKTOR < BUKSZE < BUKIZB) in case of Hosszúréti Stream and Bükkös Stream also, however CVA-plot for scale shape not support this finding. In the case of gibel carp body shape, all sampling points differed significantly. In the case of gibel carp scale shape, there was a connection with stream, but there are similarities between the sampling points from different streams as well (Fig. 5); significant differences and *Pd*-values are shown in Table 8 for body shape and Table 9 for scale shape. Interesting pattern of sites could be observed in case of gibel carp body shape, since within-stream difference (GERTOR–GERCEG) seems to be higher than between-stream (GERTOR–SZEBIC; GERTOR–HOSKEL) difference. Regarding gibel carp scale, GERTOR site have not been differed such harshly from others, like in case of body shape. BENBIA proved to be the most different site along CV1.

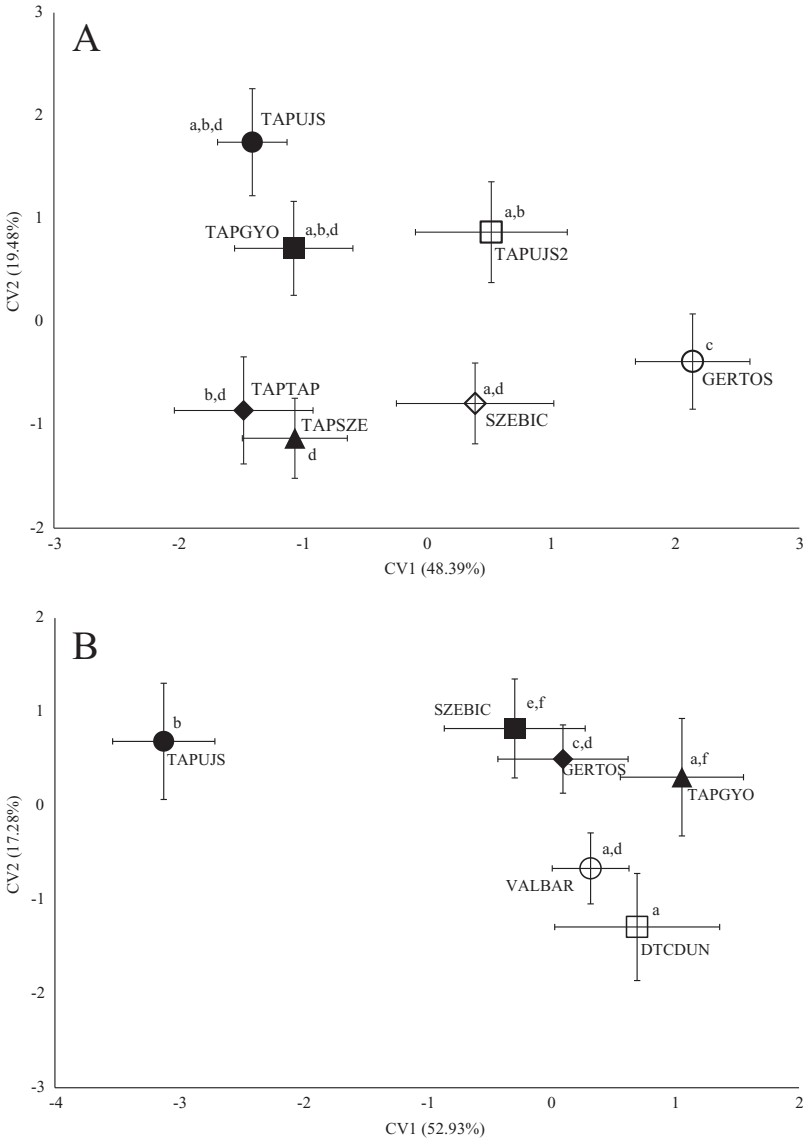

**Figure 3 Canonical Variates Analysis (CVA) results of roach (*Rutilus rutilus*) body shape (A) and scale shape (B).** Small-case letters indicate significant differences based on Procrustes distances, upper-case letters indicate the sampling points (first three letters indicates the stream). Symbols show the group centroids, crosshairs show the standard deviations.

## Significant background variables

Numerous significant background variables were found, which affect fish body shape and scale shape. Local- and landscape-scale environmental variables, water chemistry data and also PhACs were found to be significant. In case of roach scale shape, the significant variables were As (9%) and $SO_4^{2-}$ (3%), and for body shape, TRIM (6%) and CITA (4%) were found to be significant (1% joint effect). In the case of chub scale shape, water chemistry data (significant variables: Mg, As, Ca) was responsible for 5% of the variance,

Table 4 Procrustes distances (*Pd*) and *p*-values of Canonical Variates Analysis on roach (*Rutilus rutilus*) body shape.

|  |  | *p*-Values | | | | | | |
|---|---|---|---|---|---|---|---|---|
|  |  | GERTOS | SZEBIC | TAPTAP | TAPUJS | TAPGYO | TAPSZE | TAPUJS2 |
| *Pd* | GERTOS |  | **0.011** | **0.0003** | **0.0456** | **0.0074** | **0.0387** | **0.0337** |
|  | SZEBIC | 0.0353 |  | **0.0216** | 0.1186 | 0.0803 | 0.1031 | 0.7363 |
|  | TAPTAP | 0.0358 | 0.0302 |  | 0.1444 | 0.1225 | 0.7136 | **0.0269** |
|  | TAPUJS | 0.0372 | 0.0305 | 0.0288 |  | 0.5425 | 0.4209 | 0.6972 |
|  | TAPGYO | 0.0302 | 0.0218 | 0.0197 | 0.0181 |  | 0.6884 | 0.5946 |
|  | TAPSZE | 0.0308 | 0.0235 | 0.0138 | 0.0233 | 0.0131 |  | 0.3427 |
|  | TAPUJS2 | 0.0298 | 0.015 | **0.0284** | 0.02 | 0.0149 | 0.0213 |  |

Note:
Significant differences are in bold.

Table 5 Procrustes distances (*Pd*) and *p*-values of Canonical Variates Analysis on roach (*Rutilus rutilus*) scale shape.

|  |  | *p*-Values | | | | | |
|---|---|---|---|---|---|---|---|
|  |  | DTCDUN | GERTOS | SZEBIC | TAPUJS | TAPGYO | VALBAR |
| *Pd* | DTCDUN |  | **0.0213** | **0.0166** | **0.0012** | 0.4392 | 0.3091 |
|  | GERTOS | 0.0408 |  | **0.0495** | **<0.0001** | **0.0309** | 0.0639 |
|  | SZEBIC | 0.0576 | 0.0344 |  | **0.0378** | 0.1134 | **0.044** |
|  | TAPUJS | **0.0985** | **0.0753** | **0.051** |  | **<0.0001** | **0.0006** |
|  | TAPGYO | 0.0289 | **0.0332** | 0.036 | **0.0819** |  | 0.4209 |
|  | VALBAR | 0.0323 | 0.0312 | **0.0432** | **0.0862** | 0.0243 |  |

Note:
Significant differences are in bold.

local environmental variables (significant variables: emergent macrophytes, water depth) were responsible for 2% of the variance, while PhACs (significant variable: CODE) were responsible for 1% of the variance. The local environmental variables and CODE had 1% joint effect. In the case of chub body shape, only two variables were significant, Cd as water chemistry data and detritus as a local environmental variable, for 4% and 3% respectively, with 8% joint effect. In the case of gibel carp scale shape, the water chemistry variable Pb (2%) and the landscape scale environmental variable wetland (6%) were significant, with 1% joint effect. For gibel carp body shape, three different type of variables were significant, the PCB PROP, the water chemistry variable Zn, and the landscape-scale environmental variable catchment size, for 6%, 11% and 2% respectively, with 4% joint effect for Zn and catchment size (Table 10).

Mantel tests did not show significant correlation among the site-specific shape variables and the significant background variables, in most of the cases (Table S3). In case of chub scale, Ca shows significant correlation with Procrustes distances, although in case of Mahalanobis distances the correlation was not significant. In case of roach scale, both As and $SO_4^{2-}$ showed significant correlation with Mahalanobis distances, although in case of Procrustes distances the correlation was not significant.

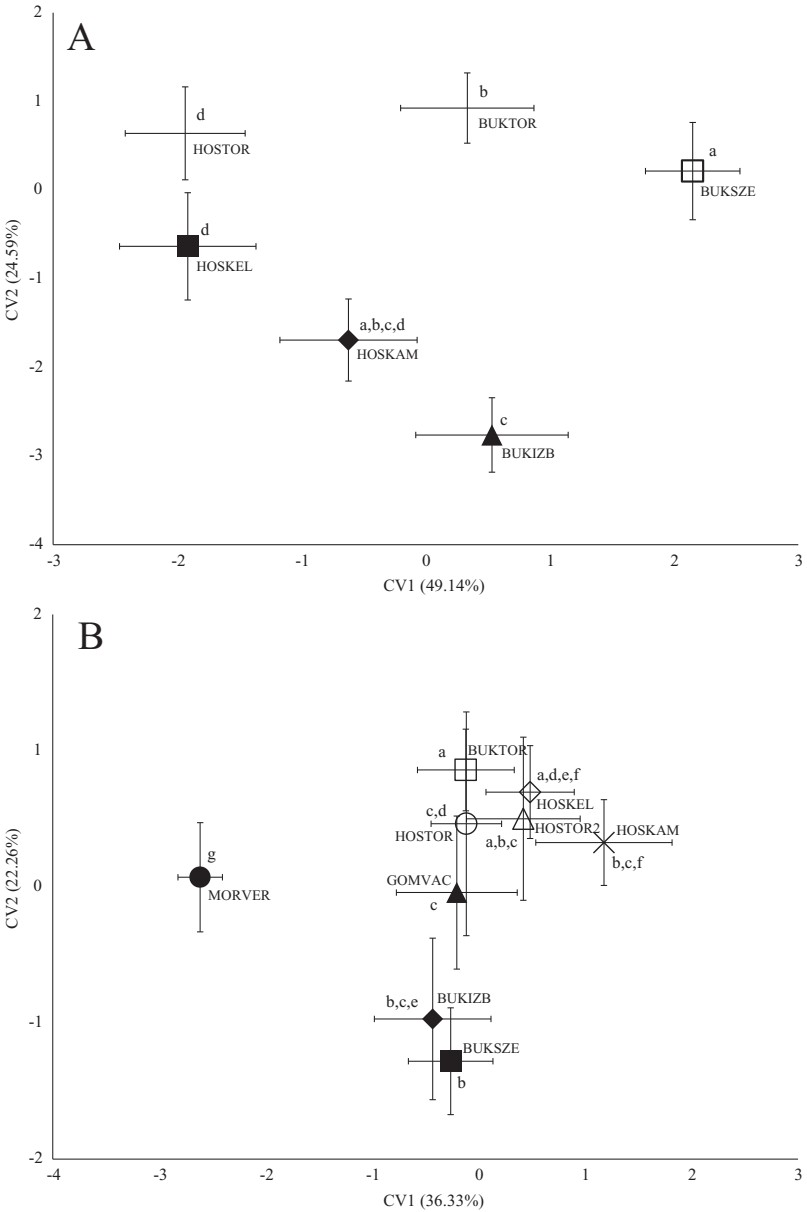

**Figure 4 Canonical Variates Analysis (CVA) results of chub (*Squalius cephalus*) body shape (A) and scale shape (B).** Small-case letters indicate significant differences based on Procrustes distances, upper-case letters indicate the sampling points (first three letters indicates the stream). Symbols show the group centroids, crosshairs show the standard deviations.

## DISCUSSION

Our results indicated that PhACs can influence fish body shape and scale shape in natural environment and habitats. There are several studies that showed shape differences between fish stocks in natural waters (*Ibáñez & Jawad, 2018*; *Takács et al., 2016*). These studies usually explain the variations by different genetic background (*Lõhmus et al., 2010*; *Staszny et al., 2013*), phenotypic plasticity (*Vasconcellos et al., 2008*), or some basic environmental differences, such as food availability (*Currens et al., 1989*; *Marcil, Swain &*

**Table 6 Procrustes distances (*Pd*) and *p*-values of Canonical Variates Analysis on chub (*Squalius cephalus*) body shape.**

| | | *p*-Values | | | | | |
|---|---|---|---|---|---|---|---|
| | | BUKIZB | BUKSZE | BUKTOR | HOSKAM | HOSKEL | HOSTOR |
| *Pd* | BUKIZB | | **0.0051** | **0.0052** | 0.2253 | **0.0441** | **0.0226** |
| | BUKSZE | 0.0292 | | **0.0046** | 0.085 | **0.0001** | **<0.0001** |
| | BUKTOR | 0.0285 | 0.018 | | 0.1404 | **0.0014** | **0.0006** |
| | HOSKAM | 0.0254 | 0.0235 | 0.021 | | 0.2441 | 0.149 |
| | HOSKEL | 0.0255 | 0.0361 | 0.0258 | 0.023 | | 0.374 |
| | HOSTOR | 0.0253 | 0.0347 | 0.0237 | 0.0238 | 0.0135 | |

Note:
Significant differences are in bold.

**Table 7 Procrustes distances (*Pd*) and *p*-Values of Canonical Variates Analysis on chub (*Squalius cephalus*) scale shape.**

| | | *p*-Values | | | | | | | | |
|---|---|---|---|---|---|---|---|---|---|---|
| | | BUKIZB | BUKSZE | BUKTOR | GOMVAC | HOSTOR | HOSKAM | HOSKEL | HOSTOR2 | MORVER |
| *Pd* | BUKIZB | | 0.8553 | **0.0092** | 0.1659 | **0.0431** | **0.0417** | 0.0673 | 0.6136 | **0.0007** |
| | BUKSZE | 0.018 | | **0.0001** | **0.0128** | **0.0002** | 0.0552 | **0.0018** | 0.085 | **0.0028** |
| | BUKTOR | **0.0426** | 0.0505 | | **0.0106** | **0.0017** | **0.021** | 0.219 | 0.3458 | **0.0004** |
| | GOMVAC | 0.0362 | **0.0376** | 0.039 | | **0.0003** | 0.1365 | **0.0293** | 0.1222 | **0.0265** |
| | HOSTOR | **0.0433** | **0.0501** | **0.0468** | 0.057 | | 0.459 | 0.549 | 0.5931 | **0.0001** |
| | HOSKAM | **0.0523** | 0.051 | **0.0574** | 0.0512 | 0.0369 | | 0.4479 | 0.3201 | **0.0237** |
| | HOSKEL | 0.0378 | **0.0443** | 0.0264 | **0.0417** | 0.0242 | 0.0359 | | 0.8486 | **0.0003** |
| | HOSTOR2 | 0.0347 | 0.0462 | 0.0342 | 0.0505 | 0.0338 | 0.0524 | 0.0271 | | **0.0068** |
| | MORVER | **0.068** | **0.069** | **0.079** | **0.0617** | **0.0882** | **0.0917** | **0.0835** | **0.0914** | |

Note:
Significant differences are in bold.

*Hutchings, 2006*; *Park et al., 2001*), temperature (*Lõhmus et al., 2010*; *Šumer et al., 2005*), flow-regime (*Haas, Blum & Heins, 2010*). These effects, and their combination have also affected the phenotype of fish included this study. Moreover, the observed impact of PhACs on shape is considered to relatively small, however it should be taken into consideration during the studies, carried out in natural waters. In addition, the results of this study suggest that the mixtures of PhACs that occur in natural waters have different effects on different species and phenotypes such as body and scale.

## Potential effects of environmental variables on shape

In the case of chub and gibel carp, significant environmental variables were found. The effects of local (section) level variables on chub scale shape could be explained by the life-history characteristics of the species. Different environmental characteristics of the given habitats may cause changes at the population level (*Haas, Blum & Heins, 2010*). Coverage of emergent macrophytes, water depth and the quantity of detritus were previously found to be connected to the life history parameters of chub (*Bolland, Cowx & Lucas, 2008*; *Ünver & Erk'akan, 2011*), therefore these variables might affect the scale and

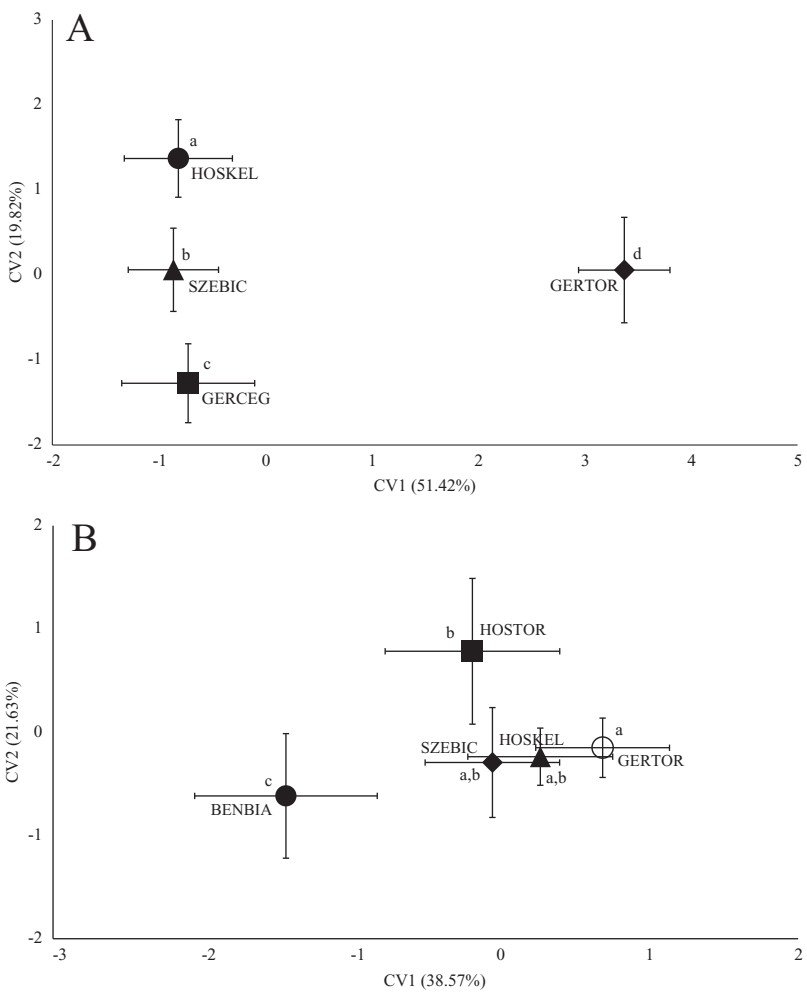

**Figure 5 Canonical Variates Analysis (CVA) results of gibel carp (*Carassius gibelio*) body shape (A) and scale shape (B).** Small-case letters indicate significant differences based on Procrustes distances, upper-case letters indicate the sampling points (first three letters indicates the stream). Symbols show the group centroids, crosshairs show the standard deviations.

body shape of the fish. In the case of gibel carp, significant environmental variables included landscape-scale variables, wetland (scale shape) and catchment size (body shape). There are several known examples regarding the shape-modification effects of environmental differences in fish. Species of the genus *Carassius* are characterized by a high level of phenotypic plasticity. In the case of crucian carp (*Carassius carassius*), the presence or absence of predators and the feeding behavior (zooplankton versus benthic chironomids) have a complex effect on body shape (*Andersson, Johansson & Söderlund, 2006*).

## Potential effects of general water chemistry on scale shape

Water chemistry had a significant impact on roach and chub scale shape. The effects of arsenic (As) on muscle development in fish have already been reported (*D'Amico, 2012*), and this compound can accumulated in scales (*Allen, Awasthi & Rana, 2004*) as well, which

**Table 8 Procrustes distances (*Pd*) and *p*-values of Canonical Variates Analysis on gibel carp (*Carassius gibelio*) body shape.**

|  |  | *p*-Values | | | |
|---|---|---|---|---|---|
|  |  | GERCEG | GERTOR | HOSKEL | SZEBIC |
| *Pd* | GERCEG |  | **<0.0001** | **0.0047** | **<0.0001** |
|  | GERTOR | 0.036 |  | **<0.0001** | **<0.0001** |
|  | HOSKEL | 0.0261 | 0.0438 |  | **<0.0001** |
|  | SZEBIC | 0.0247 | 0.0475 | 0.0441 |  |

Note:
Significant differences are in bold.

**Table 9 Procrustes distances (*Pd*) and *p*-values of Canonical Variates Analysis on gibel carp (*Carassius gibelio*) scale shape.**

|  |  | *p*-Values | | | | |
|---|---|---|---|---|---|---|
|  |  | BENBIA | GERTOR | HOSKEL | HOSTOR | SZEBIC |
| *Pd* | BENBIA |  | **0.0002** | **0.0175** | **0.0137** | **0.0038** |
|  | GERTOR | 0.0676 |  | 0.111 | **0.0229** | 0.3999 |
|  | HOSKEL | 0.0601 | 0.0428 |  | 0.5836 | 0.2426 |
|  | HOSTOR | 0.0534 | 0.0475 | 0.0346 |  | 0.0854 |
|  | SZEBIC | 0.0504 | 0.0246 | 0.037 | 0.038 |  |

Note:
Significant differences are in bold.

**Table 10 Proportion of significant background variables on fish body shape and scale shape.**

| Species | Analyzed shape | Variable category | Significant variable | Proportion of effect (%) | Joint effect (%) |
|---|---|---|---|---|---|
| Roach | Scale | C | As | 9 |  |
|  |  | C | $SO_4^{2-}$ | 3 |  |
|  | Body | PhAC | TRIM | 6 | 1 |
|  |  | PhAC | CITA | 4 |  |
| Chub | Scale | C | Mg | 5 |  |
|  |  | C | As |  |  |
|  |  | C | Ca |  |  |
|  |  | LE | Macrophyte coverage | 2 | 1 |
|  |  | LE | Water depth |  |  |
|  |  | PhAC | CODE | 1 |  |
|  | Body | LE | Detritus | 3 | 8 |
|  |  | C | Cd | 4 |  |
| Gibel carp | Scale | LSE | Wetland | 6 | 1 |
|  |  | C | Pb | 2 |  |
|  | Body | C | Zn | 11 | 4 |
|  |  | LSE | Catchment size | 2 |  |
|  |  | PhAC | PROP | 6 |  |

Note:
Variable types: C, water chemistry data; PhAC, pharmaceutical active compound; LE, local environmental variables; LSE, landscape scale environmental variables.

might affect scale shape itself. *Fliedner et al. (2014)* studied the water chemistry, especially the heavy metal concentrations in rivers Rhine, Elbe, Danube, Saar, Mulde, Saale and in Lake Belau in Germany. Throughout the study As, Pb, Cu and Hg concentrations were measured from tissue samples of zebra mussel (*Dreissena polymorpha*) and bream (*Abramis brama*). Arsenic found to be the only compound, where increase in concentration was detectable while analyzing in bream muscle tissue samples from 1990s to 2014 (*Fliedner et al., 2014*). $Mg^{2+}$ and $Ca^{2+}$ significantly impacted the scale shape of chub. $Ca^{2+}$ is an essential building component of fish scales (*Sankar et al., 2008*) while the $Mg^{2+}$ content of water affects calcium uptake in fish (*Dabrowska, Meyer-Burgdorff & Gunther, 1991*; *Van der Velden et al., 1991*). Cadmium is a $Ca^{2+}$ uptake inhibiting agent which was also shown to affect chub body shape. The presence of Cd has a negative effect on $Ca^{2+}$ uptake through the gills (*Franklin et al., 2005*). Lead concentrations are also connected to gibel carp scale shape formation. This heavy metal cannot be excreted physiologically (via the gills or kidneys), and Pb impairs fish scale development to a greater extent than in other organs (*Çoban et al., 2013*). Zinc also has a significant impact on gibel carp body shape, and is associated with higher (11%) variance. Zinc uptake is related to $Ca^{2+}$ concentrations where high $Ca^{2+}$ concentrations may decrease Zn uptake; excess Zn then accumulates in fish skin, muscle and bones (*Hogstrand & Wood, 1996*), and therefore might have an effect on body shape.

## Potential effect of PhACs on shape

TRIM is a cytoprotective, anti-ischemic agent with a strong antioxidant effect (*Sedky et al., 2017*). In zebrafish (*Danio rerio*) TRIM can decrease the ototoxic effects of neomycin on hair-cell loss in the neuromasts (*Chang et al., 2013*). Phenotypic alterations have not been discussed previously, however, a significant effect was detected on roach body shape in this study. CITA as a SSRI, have also been shown to significantly affect roach body shape. A strong anxiolytic effect has been reported in fish previously (*Olsén et al., 2014*; *Porseryd et al., 2017*), and alterations in behavioral patterns might also affect the phenotype as well, because the use of different habitats might alter the phenotype of different species (*Faulks et al., 2015*). CODE an opiate derivative, is used to treat rheumatic pain (*Ytterberg, Mahowald & Woods, 1998*), and significantly modulates chub-scale shape. There is evidence of the presence of codeine in fish tissues (*Epple et al., 1993*; *Valdés et al., 2016*), however, phenotypic alterations have not been detected. It might be in relation with the inhibition of the expression of receptors for vascular endothelial growth factor, which can affect the early life-stage development of fish (*Karaman et al., 2017*). PROP, a non-selective β-blocker, affected gibel carp body shape. It is used to treat heart diseases, and has proved to be the cause of decreased testosterone and estradiol levels in zebrafish, and has showed anxiolytic effects, and decreased growth (*Mitchell & Moon, 2016*). As we discussed in the case of roach and CITA, the anxiolytic effects of drugs might also alter phenotype. Based on RQ-values, CITA was ranked to be high risk, while CODE and PROP were medium risk and TRIM was low risk. These results also suggest that the widely used "traditional" risk assessment may have weaknesses when compared to a "real-life" measured effects.

## CONCLUSIONS

In summary, our results suggest that PhACs in natural waters can affect the phenotypic characteristics of fish species. Although a relatively large number of PhACs (54 compounds) were found in the water samples, only 4 compounds were found to have significant effects on phenotype. This study did not aim to find clear cause and effect relationships between the given compounds, or to reveal the mode-of-actions; however, the individual-scale effect of PhACs was identified. The results of this study showed that differences in phenotype can be detected, therefore the morphometric analysis was suitable for an alternative, sub-lethal endpoint of environment-level toxicological investigation. However, in order to get a more accurate picture of the actual phenotypic effect of PhACs in the environment, a more detailed study with a larger sample size is needed. Since the effects of PhACs on scale shape have been observed, scale sampling may be a suitable, effective and ethically acceptable tool to extend studies on different river systems.

## ACKNOWLEDGEMENTS

The authors thank Andrew J. Hamer (University of Melbourne, ELKH Balaton Limnological Institute) for the English proofreading, Petra Dragan (Hungarian University of Agriculture and Life Sciences) and Igor Dukay (Renatur Bt.) for defining the WWTPs location.

### Funding

The research was financed by the National Research, Development and Innovation Office (NKFIH) from the NRDI Fund, Hungary. Identification number: NVKP_16-1-2016-0003. Adam Staszny and Gabor Maasz was supported by the Bolyai Fellowship of the Hungarian Academy of Sciences (BO/00407/17/4 Adam Staszny, BO/000549/20/7 Gabor Maasz) and the (ÚNKP-19-4 Adam Staszny, ÚNKP-20-5 Gabor Maasz) New National Excellence Program of the Ministry for Innovation and Technology, Hungary. The publication is supported by the EFOP-3.6.3-VEKOP-16-2017-00008 project. Arpad Ferincz, Vera Juhasz and Bela Urbanyi has been supported by the NKFIH-831-10/2019 and the TKP2020-NKA projects. The chemical analysis of PhACs and the environmental risk assesment was supported by the National Brain Project, Hungary (No. 2017-1.2.1-NKP-2017-00002). The funders had no role in study design, data collection and analysis, decision to publish, or preparation of the manuscript.

### Grant Disclosures

The following grant information was disclosed by the authors:
NRDI Fund, Hungary: NVKP_16-1-2016-0003.
Hungarian Academy of Sciences: BO/00407/17/4 and BO/000549/20/7.
Ministry for Innovation and Technology, Hungary: ÚNKP-19-4 and ÚNKP-20-5.

EFOP-3.6.3-VEKOP-16-2017-00008.
NKFIH-831-10/2019.
National Brain Project: 2017-1.2.1-NKP-2017-00002.

## Competing Interests

The authors declare that they have no competing interests.

## Author Contributions

- Adam Staszny performed the experiments, analyzed the data, prepared figures and/or tables, authored or reviewed drafts of the paper, and approved the final draft.
- Peter Dobosy performed the experiments, authored or reviewed drafts of the paper, and approved the final draft.
- Gabor Maasz performed the experiments, analyzed the data, authored or reviewed drafts of the paper, and approved the final draft.
- Zoltan Szalai analyzed the data, authored or reviewed drafts of the paper, and approved the final draft.
- Gergely Jakab analyzed the data, authored or reviewed drafts of the paper, and approved the final draft.
- Zsolt Pirger performed the experiments, analyzed the data, authored or reviewed drafts of the paper, and approved the final draft.
- Jozsef Szeberenyi analyzed the data, prepared figures and/or tables, authored or reviewed drafts of the paper, and approved the final draft.
- Eva Molnar performed the experiments, analyzed the data, prepared figures and/or tables, authored or reviewed drafts of the paper, and approved the final draft.
- Lilianna Olimpia Pap performed the experiments, authored or reviewed drafts of the paper, and approved the final draft.
- Vera Juhasz performed the experiments, authored or reviewed drafts of the paper, and approved the final draft.
- Andras Weiperth performed the experiments, authored or reviewed drafts of the paper, and approved the final draft.
- Bela Urbanyi conceived and designed the experiments, authored or reviewed drafts of the paper, and approved the final draft.
- Attila Csaba Kondor conceived and designed the experiments, authored or reviewed drafts of the paper, and approved the final draft.
- Arpad Ferincz conceived and designed the experiments, performed the experiments, analyzed the data, prepared figures and/or tables, authored or reviewed drafts of the paper, and approved the final draft.

## Animal Ethics

The following information was supplied relating to ethical approvals (i.e., approving body and any reference numbers):

Fish collection for laboratory examinations was authorized by the Government Office of Pest county (Permit no.: XIV-I-001/2302-4/2012).

## Field Study Permissions

The following information was supplied relating to field study approvals (i.e., approving body and any reference numbers):

Fish samplings have been authorized by the Minister of Agriculture (Permit no.: HHgF/ 298-1/2016).

## Data Availability

Raw data, including risk assessment data, Canonical Variates Analysis results for roach, chub and gibel carp body and scale shape, and the measured PhACs in the water samples, are available in the Supplemental Files.

## Supplemental Information

Supplemental information for this article can be found online at http://dx.doi.org/10.7717/ peerj.10642#supplemental-information.

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
