# Peer review of "Effects of pharmaceutically active compounds (PhACs) on fish body and scale shape in natural waters"

_PeerJ, doi:10.7717/peerj.10642_

## Round 0.1 · original submission · Major Revisions

The reviewers highlight the strengths of your submitted manuscript and have proposed a number of changes to improve the quality. Please consider the proposed changes as mandatory for final acceptance of your manuscript.

While some comments address more technical aspects such as the improvement of language and of statistical procedures (reviewer 1), there are also content-related aspects to be resolved. These include a more detailed introduction, additional information on sampled fish, limitations of your analyses due to probably biased sex ratios or unequal age classes among stations (mainly mentioned by reviewer 2).

I look forward to your revised manuscript.

·

Basic reporting

The manuscript provides interesting data and analyses that clearly fit the standards of the journal. The quality of the English language seems to be high, however, I am not a native speaker.

Experimental design

Data collection and data analyses are well designed. Research questions are well defined with an identified knowledge gap. Investigations were performed in the high technical and ethical standard.

Validity of the findings

In the first view, the analyses looked a little bit like the data dredging, but the explained possible mechanism how pharmaceutically active compounds can influence the fish body shape or/and scale shape eliminated this feeling. Conclusions are well stated. I have one issue related to the statistical analyses. Authors collected 20 fish specimen per site, so data are spatially replicated, so there could be not the real effect of a significant variable. E.g. just speculation, water velocity can influence body shape, but it also can more intensively flush away the compound. I would suggest including analyses where this issue can be eliminated. Is it possible to test this effect by some mixed models that can eliminate site-specific patterns, e.g. just compare the body and scale shape among sites using some alternative of PERMANOVA or interaction between variables? In the case of non-significant results, the changes in body or scale shape can be caused by pharmaceutically active compounds.

Additional comments

The overuse of abbreviations, e.g. abbreviation of pharmaceutically active compounds is quite disturbing during the reading, especially when the reader is not familiar with the abbreviations (like me). I would suggest changing these abbreviations with full names.

Reviewer 2 ·

Basic reporting

Overall, the paper is well written, but there are a few sentences where the English could be edited to improve readability (e.g. lines 66, 207, 273).

References seem appropriate

Article structure is professional and raw data are provided.

Experimental design

This is original primary research with a well defined and approached research question.

Methods are described with sufficient detail & information to replicate.

Validity of the findings

see general comments

Additional comments

In their manuscript, Stazny et al investigate whether exposure to PhACs could possibly play a role in altering the phenotypic characteristics scale and body morphology of three low land fish species. To this extend river water samples were analyzed from the sites where fish collections took place. The water analyses are extensive comprising general water parameters, ion measurements as well as a suite of PhACs. The authors provide references describing that, to date, certain water chemistry parameters are known to have effects on body and scale shapes whereas little is known to this effect for PhACs.
The approach taken is interesting and if thoroughly validated, could certainly be a valuable addition to existing risk assessment approaches. It is a shame that there is not a single river where the authors could catch all three species which would really have strengthened the paper.
Whilst the discussion has an appropriate length, I feel the introduction could benefit from a bit more detail (e.g. a brief description on current knowledge of effects of water chemist parameters on phenotypes which is discussed in detail in the discussion. Overall, the paper is well written, but there are a few sentences where the English could be edited to improve readability (e.g. lines 66, 207, 273).

Specific comments:
I am missing some information on the age groups and sex (it has been determined, line 158) of the sampled fish as both measured endpoints (body and scale shape) could possibly differ at different age or sex. The latter is especially the case depending on when sampling occurred as if fish were sampled during the spawning season, females are definitely rounder in body shape compared to times outside of the breading season.
Are any sampling sites considered as controls, i.e. not impacted by the presence of PhACs? A classic approach would be sampling upstream and downstream of WWTWs as described for instance by Jobling et al, Env Health Persp 2006 or Tyler & Jobling, BioScience 2008). As there were different sampling sites along different rivers, it might be the case that fish were sampled up- and downstream of WWTWs, but at the moment this is not clear in the paper.

Was a comparison of body/scale shape performed within one species across the different sampling sites to exclude possible population differences? A few words to the effect would be beneficial in the paper.

Lines 39- 40 Although E1, E2, EE2 and E3 are common abbreviations, they should be defined at first mention. Also, what is aE2 (seeing that data for bE2 are also provided in table 3, this cannot me a typo)

Line 47 be more specific where the first PhACs where detected. It is probably meant in aquatic ecosystems, the aquatic environment or similar.

Lines 83 – 84 Be more specific with regards to ‘relevant national and international guidelines concerning the care and welfare of fish’. This is a very generic statement and should be supported by references.

Lines 101-135 The methods description for chemical analysis is sufficiently detailed which is nice to see, but I am missing the mention of limits of detection for the different variables measured, especially the PhACs (this could even be included in Table 3.

Line 158 I would question the accuracy of wet weight determined after freezing and defrosting. It has been shown that post-thaw wet weight differed from pre-storage wet weight (Crane et al 2016, Fisheries Research). Has any correcting factor been applied?

Line 164/Figure2 Are scale and body shape of roach and chub comparable to gibel carp? It would be helpful for the reader to show images for all three species in figure 2.

Lines 182-187 It is not entirely clear how the PNECs and especially the assessment factor AF were derived.

Lines 191-192 the log10 transformation is unclear. Also, how can this be applied to landscape variables which are not numerical if I understand correctly.

Lines 213-222 There could be a little more detail/description explaining the results presented in the corresponding tables.

Lines 213-222 The results described here refer to figures which are informative, but these figures only show certain sampling sites for each species and not data for all sites where the corresponding species were sampled. What is the reason for omitting some sampling sites?
Lines 229-234 The overall largest joint effect presented in table 10 is for chub on body shape (8%), but is not reported at all in this results section whilst joint effects of 1%? On roach and gibel carp are. Please add or explain why this has been omitted.

Lines 242-249 It is good to see a cautious statement on possible effects of PhACs on body and scale shapes of wild fish and a mention of possible underlying reasons. I would argue that there are a probably a variety of other factors too, including genetics and adaptation that, over years/decades, could have led to slight differences in body shape of different, independent populations of the same species inhabiting different rivers/streams that are not connected and that PhACs play a minor role in this. As mentioned above, phenotypic plasticity, genetics and adaptation are the most likely factors underlying the differences in shape. Therefore, as suggested, it would be beneficial, to also draw a comparison between fish of the same species collected at different river systems.

Lines 270-290 Nice discussion how certain ions/PhACs could result in different body/scale shape and nicely supported by relevant literature.

Lines 323-325 Good conclusion making the point that more studies of this nature and with larger sample sizes are required. A critical point to be added here though would be of ethical nature, i.e. the use of too many animals. Therefore, a comparison between different river systems as suggested might be a first step.

Table 3 Is there a reason that some compounds in table 3 are written in bold, others in italics and others in normal? If there is a reason, this needs ort be explained in the table legend.

Table 3 What is the difference between aE2 and bE2? This needs to be defined.

---

## Round 0.2 · Minor Revisions

Thank you very much for the revision of the paper which generally looks fine now. However, the Abstract is a bit too abstract as it currently stands.

Please be more specific and more orientated to the results you have obtained. For example, lines 39-41: it would seem to be good to add exactly what (kind/style) risk levels we are dealing with here, e.g. "government-delineated risk levels", or "risk of XXXX" etc. Overall, the abstract needs to be easier to understand.

Reviewer 2 ·

Basic reporting

Interesting data that are well presented in relation to existing literature.

Experimental design

The experiment is well designed and unclarities have been well addressed in the revised version.
The only small criticism would be that that age of the fish has not been determined. Having scales to hand would have easily allowed to determine the age of the sampled fish which might have been a useful factor in the statistical analyses, especially seeing the size variation for roach and giebel carp reported in table S1. However, the process is time consuming and possibly not feasible in the time the revision was due.

Validity of the findings

Interesting data and a novel approach that can add to existing risk assessment approaches. Raw data have all been provided.

Additional comments

More out of interest, in the revised version, it is mentioned that the scale shape of roach at one sampling site differed from most other sites. Do the authors have any idea why this could be? IT is well known that roach can interbreed with other cyprinid species such as rudd (Scardinius erytrophthalmus) or bream (Abramis brama) (Wyatt et al., 2006, J. Fish Biol., 69, 52–71. https://doi.org/10.1111/j.1095‐8649.2006.01104.x) and it woudl be interesting if any of these species inhabited this particular stream?

---

## Round 0.3 · accepted · Accept

Thank you for updating the abstract which is now improved.